# TGF-β Type I Receptor Signaling in Melanoma Liver Metastases Increases Metastatic Outgrowth

**DOI:** 10.3390/ijms24108676

**Published:** 2023-05-12

**Authors:** Dieuwke L. Marvin, Jelmer Dijkstra, Rabia M. Zulfiqar, Michiel Vermeulen, Peter ten Dijke, Laila Ritsma

**Affiliations:** 1Oncode Institute and Department of Cell and Chemical Biology, Leiden University Medical Center, 2333 ZC Leiden, The Netherlands; 2Oncode Institute and Radboud Institute for Molecular Life Sciences (RIMLS), Radboud University, 6525 GA Nijmegen, The Netherlands

**Keywords:** TGF-β signaling, melanoma, liver metastasis, tumor microenvironment

## Abstract

**Simple Summary:**

Melanoma patients with liver metastasis have a poor prognosis, despite progress in therapeutic strategies. The cytokine Transforming Growth Factor β (TGF-β) plays a role in melanoma cells and acts on cells in the liver. We hypothesized that this cytokine influences the metastatic outgrowth of melanoma in liver. To investigate this, we generated a model to turn on and off TGF-β signaling in the B16F10 melanoma cells. In vitro, TGF-β activation repressed B16F10 melanoma cell growth and migration, while in vivo, sustained TGF-β activation increased outgrowth in liver. In the tumor microenvironment, TGF-β activation led to changes in immune cells. Analysis of newly secreted proteins revealed that B16F10 cells in which the TGF-β pathway is overly active secrete more matrix remodeling proteins. Such proteins that surround cells could directly or indirectly lead to changes in immune cell compartments during liver metastasis. Our results contribute to the understanding of TGF-β in liver metastasis of melanoma cells.

**Abstract:**

Despite advances in treatment for metastatic melanoma patients, patients with liver metastasis have an unfavorable prognosis. A better understanding of the development of liver metastasis is needed. The multifunctional cytokine Transforming Growth Factor β (TGF-β) plays various roles in melanoma tumors and metastasis, affecting both tumor cells and cells from the surrounding tumor microenvironment. To study the role of TGF-β in melanoma liver metastasis, we created a model to activate or repress the TGF-β receptor pathway in vitro and in vivo in an inducible manner. For this, we engineered B16F10 melanoma cells to have inducible ectopic expression of a constitutively active (ca) or kinase-inactive (ki) TGF-β receptor I, also termed activin receptor-like kinase (ALK5). In vitro, stimulation with TGF-β signaling and ectopic caALK5 expression reduced B16F10 cell proliferation and migration. Contrasting results were found in vivo; sustained caALK5 expression in B16F10 cells in vivo increased the metastatic outgrowth in liver. Blocking microenvironmental TGF-β did not affect metastatic liver outgrowth of both control and caALK5 expressing B16F10 cells. Upon characterizing the tumor microenvironment of control and caALk5 expressing B16F10 tumors, we observed reduced (cytotoxic) T cell presence and infiltration, as well as an increase in bone marrow-derived macrophages in caALK5 expressing B16F10 tumors. This suggests that caALK5 expression in B16F10 cells induces changes in the tumor microenvironment. A comparison of newly synthesized secreted proteins upon caALK5 expression by B16F10 cells revealed increased secretion of matrix remodeling proteins. Our results show that TGF-β receptor activation in B16F10 melanoma cells can increase metastatic outgrowth in liver in vivo, possibly through remodeling of the tumor microenvironment leading to altered infiltration of immune cells. These results provide insights in the role of TGF-β signaling in B16F10 liver metastasis and could have implications regarding the use of TGF-β inhibitors for the treatment of melanoma patients with liver metastasis.

## 1. Introduction

Despite the introduction of immune checkpoint inhibitors and BRAF/MEK inhibitors, the prognosis of patients with metastatic melanoma remains unfavorable. In particular, patients with liver metastasis show a poor response to current therapies resulting in a poor prognosis [1,2,3,4,5]. Further studies into the mechanisms of liver metastasis are therefore necessary to better understand the development of liver metastasis and improve therapeutic response.

Transforming Growth Factor (TGF)-β is a multifunctional cytokine involved in various processes in both healthy and tumor cells, and can have a pro-metastatic role [6,7]. TGF-β cytokines are part of the TGF-β family, which includes the three TGF-β isoforms TGF-β1, TGF-β2, and TGF-β3, and other structurally and functionally related proteins. TGF-β is secreted in a latent form in which the latency-associated N-terminal precursor shields the mature bioactive part from interacting with the cell surface transmembrane TGF-β receptors; activation of latent TGF-β occurs upon interaction with cell surface integrins [6]. TGF-β signaling starts by binding to the TGF-β serine/threonine kinase type II receptor (TβRII), followed by heteromeric complex formation and phosphorylation of TGF-β type I receptor (TβRI), also termed activin receptor-like kinase-5 (ALK5). SMAD 2/3 proteins are phosphorylated by the activated TβRI and form heteromeric complexes with SMAD4. The heteromeric SMAD complexes translocate to the nucleus to regulate the expression of TGF-β target genes. The negative feedback of the pathway is regulated by, amongst other factors, the expression of the TGF-β/SMAD target gene *SMAD7*, which together with SMURF2 E3 ubiquitin ligase induces the ubiquitin-dependent degradation of ALK5.

The effects of TGF-β signaling are highly cell and context specific [6]. In tumor cells, TGF-β plays a dual role, where it can both repress cell proliferation and/or promote migration and invasion [6]. In melanocytes and pre-malignant cells, canonical TGF-β signaling results in cell cycle arrest, while malignant melanoma can become resistant to its cytostatic effects [8]. Increased TGF-β levels and TGF-β signaling have been associated with melanoma progression and invasion, suggesting a pro-tumorigenic role of TGF-β in malignant melanoma cells [9,10,11,12]. In addition to tumor cells, TGF-β affects various cells present in the liver tumor microenvironment, including fibroblasts and immune cells [7,13,14,15].

Since TGF-β can contribute to pro-tumorigenic processes by acting on both tumor cells and microenvironmental cells in the liver, we hypothesize that TGF-β signaling plays a key role in the development of melanoma liver metastasis. In this study we generated a conditional model to activate and repress TGF-β signaling by inducible ectopic expression of a constitutively active (ca) or kinase-inactive (ki) ALK5. Using this model, we investigated the role of TGF-β in B16F10 melanoma cells in vitro, as well as in B16F10 liver metastasis in vivo. Our findings reveal contrasting results between TGF-β activation in vitro and in vivo. In vivo caALK5 expression increases metastatic liver outgrowth of B16F10 cells. The results of this study give insights in the role of TGF-β signaling in B16F10 liver metastasis.

## 2. Results

### 2.1. An Inducible Cellular Model to Activate and Repress TGF-β Type I Receptor Signaling

To investigate the role of TGF-β type I receptor signaling in melanoma cell invasion and metastasis, we generated a doxycycline (dox) inducible melanoma cell model to activate and repress TGF-β signaling. Constructs expressing a human constitutively active (ca) ALK5 mutant variant (ALK5^T204D^) [16] and a kinase-inactive (ki)ALK5 variant (ALK5^K232R^) [16] were inserted in lentiviral p.Inducer20 plasmids containing a dox-dependent promotor [17]. The resulting plasmids were transduced in B16F10 cells, leading to B16F10 caALK5 and B16F10 kiALK5 cells (Figure 1A). Analysis of mRNA levels revealed that expression of human *ALK5* was observed after dox treatment, while the murine *Alk5* mRNA levels remained unchanged upon dox treatment (Appendix A).

We then validated the activation and repression of TGF-β receptor signaling by examining the effect of ectopic ca/kiALK5 expression on the level of SMAD2 phosphorylation, one of the first events during TGF-β intracellular signaling (Figure 1B and Appendix A). Indeed, the phosphorylation of SMAD2 was observed upon expression of caALK5, and it was repressed upon kiALK5 expression. Next, the expression levels of TGF-β target genes, i.e., *Serpine1*, *Ctgf*, and *Smad7*, were investigated at the mRNA level (Figure 1C–E and Appendix A) and protein level (Figure 1B). Upon caALK5 expression, the gene expression of TGF-β target genes *Serpine1*, *Ctgf*, and *Smad7,* as well as the PAI-1 protein expression, increased, like TGF-β stimulation. Importantly, the caALK5-mediated effects were blocked by the treatment of cells with the small-molecule ALK5 kinase inhibitor SB505124 (Figure 1B–E and Appendix A), suggesting the effects were mediated by ALK5 kinase activity. Contrarily to caALK5, the expression of kiALK5 repressed TGF-β-induced gene and protein expression (Figure 1B–E and Appendix A).

In addition to the increased levels of the TGF-β target gene and protein expression, the expression of caALK5 resulted in the activation of the TGF-β CAGA_12_-SMAD3/4-driven transcriptional luciferase reporter to similar levels as the TGF-β treatment (Figure 1F). In contrast, kiALK5 expression repressed the activation of the TGF-β CAGA_12_-luciferase reporter upon TGF-β stimulation. We thus confirm a dox inducible model of the activation and repression of TGF-β receptor signaling in murine melanoma cells.

### 2.2. TGF-β Decreases B16F10 Cell Proliferation and Migration In Vitro

TGF-β has a dual role in cancer, being able to both repress cell proliferation as well as stimulate migration and invasion [6]. We treated B16F10 cells with TGF-β and measured cell proliferation over time (Figure 2A and Appendix A). We observed a significant reduction in cell proliferation as measured by reduced metabolic activity (Figure 2A), colony formation (Appendix A), and confluency (Appendix A). caALK5 expression resulted in a similar reduction in cell proliferation (Figure 2A), while the TGF-β-induced inhibitory effect on cell proliferation was rescued by kiALK5 expression (Figure 2A). Next, the effect of TGF-β on B16F10 migration was studied using wound healing assays under serum-starved conditions to minimize the effects of decreased proliferation on cell migration. TGF-β-treated B16F10 cells were slower in replacing the wound area compared to vehicle control-treated cells (Figure 2B). Moreover, similar effects were observed for caALK5 expressing cells, and the TGF-β-induced inhibitory effects on cell migration were rescued by kiALK5 expression. To further validate these results, we also assessed single-cell migration (Appendix A) and wound migration through matrigel (Appendix A). These results were consistent with those of other cell migration assays, in that we also observed reduced migration and invasion upon TGF-β stimulation or induced caALK5 expression. In conclusion, TGF-β or the ectopic expression of caALK5 reduced both B16F10 cell proliferation and migration in vitro.

### 2.3. Metastatic Liver Outgrowth of B16F10 Cells Is Enhanced upon TGF-β Signaling in Tumor Cells

Next, we investigated the role of TGF-β in B16F10 cells in liver metastasis in vivo. To induce experimental liver metastases, we injected clonal B16F10 cell lines containing either the dox inducible caALK5 or kiALK5 in the mesenteric vein of C57BL/6J mice [18]. Clonal cell lines were selected based on similar caALK5 and kiALK5 expression (Appendix A). To induce ALK5 expression, the mice were treated with control chow or chow containing dox (625 mg/kg) and dox treatment of cells was started one day prior to injection (Figure 3A) [18]. The induced expression of ALK5 could be detected at the protein level, although levels varied, possibly due to differential contributions of metastasis in protein samples (Appendix A). After two weeks, the metastatic outgrowth in liver was determined by calculating the percentage of metastatic replacement in liver tissue sections. We found that, upon dox treatment, B16F10 cells expressing caALK5 showed an increased metastatic liver outgrowth compared to their untreated control (Figure 3B,C). In contrast, B16F10 cells expressing kiALK5 showed decreased metastatic liver outgrowth upon dox treatment compared to the untreated control. Both the number and size of the B16F10 metastasis increased upon caALK5 expression, suggesting that ALK5 activation most likely affected both initial survival (number) and outgrowth (size) (Figure 3D,E). Indeed, when mice injected with B16F10-expressing caALK5 were treated with dox during initial seeding only (up to 24 h after injection), the effect of caALK5 expression on metastatic growth and number was eliminated (Figure 3F). We confirmed the in vivo activation of the TGF-β receptor/SMAD signaling pathway in the caALK5-expressing B16F10 liver metastases by staining for TGF-β target protein PAI-1; increased PAI-1 levels were observed in sections of metastatic lesions in caALK5 expressing B16F10 cells (Figure 3G,H).

Our data shows a disconnection between the in vivo and in vitro data; tumor cells seemed to respond differently to TGF-β in vivo compared to in vitro as TGF-β signaling in tumor cells decreased proliferation and migration in vitro, but increased metastatic outgrowth in liver in vivo.

### 2.4. Metastatic Liver Outgrowth of Control and caALK5 Expressing B16F10 Cells Is Not Affected by TGF-β Neutralizing Antibody

Since TGF-β itself can have effects on various tumor micro-environmental cells in the liver that contribute to metastasis [7] we investigated whether TGF-β in the tumor microenvironment also contributes to B16F10 liver metastasis formation. The pan-TGF-β neutralizing antibody 1D11 binds and therefore inhibits all three TGF-β isoforms from binding the receptor. However, TGF-β signaling induced by caALK5 expression should remain unchanged upon 1D11 treatment, allowing TGF-β signaling inhibition specifically in the tumor microenvironment, while keeping TGF-β signaling active in tumor cells.

We first assessed the effect of 1D11 treatment on caALK5 induced TGF-β signaling using an in vitro setup. As expected, the effects of caALK5 expression in B16F10 cells on TGF-β signaling were unaffected by 1D11 treatment (Figure 4A,B). Thus, this set-up indeed allowed us to separate the effects of TGF-β receptor activation on tumor cells (through dox induced caALK5 expression) versus microenvironmental cells (through paracrine TGF-β signaling). We then induced control and caALK5 expressing B16F10 liver metastases, and we treated mice with control IgG or 1D11 (Figure 4C). In the B16F10 liver metastasis model, 1D11 did not result in changes in metastatic liver outgrowth of B16F10 cells, in both control and caALK5 expressing cells (Figure 4D,E). We thus concluded that micro-environmentally active TGF-β does not play a role in B16F10 or TGF-β active B16F10 liver metastasis.

### 2.5. Changes in Immune Cell Presence upon caALK5 Expression in B16F10 Metastasis

caALK5 expression in B16F10 cells drastically increased metastatic liver outgrowth of B16F10 cells in the liver, which is contrasting to findings in vitro. We hypothesized this difference is due to changes in the tumor microenvironment upon caALK5 expression promoting B16F10 outgrowth, as this compartment is not present in vitro. We therefore aimed to characterize the tumor microenvironment of control and caALK5 expressing B16F10 metastasis. The tumor microenvironment comprises various recruited and resident cell types that can support the (out)growth of metastatic cells, including fibroblasts and different immune cells.

Tumor-associated fibroblasts (CAFs) can play a key role in supporting the metastatic outgrowth of many tumors, including melanoma, for example, by secreting growth factors and matrix (remodeling) proteins [19]. When staining B16F10 control and caALK5 expressing liver metastasis for the CAF marker platelet-derived growth factor receptor (PDGFR)-β, no changes in CAF staining was observed (Figure 5A). Recruited (bone marrow-derived) macrophages or resident liver macrophages (Kupffer cells) can play a role during metastatic outgrowth by, amongst other processes, providing an immune-evasive environment [20,21,22]. Staining for the total macrophage population (Iba1) revealed no difference in total macrophage numbers (Figure 5B and Appendix A). However, liver-resident C-type lectin domain family 4 (CLEC4F)^+^ macrophages seem to be reduced upon caALK5 expression in B16F10 cells (Figure 5C and Appendix A). During metastasis formation, bone marrow-derived macrophages (BMDM) can be recruited to promote an anti-inflammatory environment. We assessed the presence of BMDMs using the CCR2 marker in sections from both the control and caALK5 tumor liver metastasis. We observed an increase in CCR2^+^ macrophages in caALK5 tumors compared to control tumors (Figure 5D and Appendix A). These results suggest there could be a switch in macrophage populations mediated by the caALK5-expressing B16F10 liver metastasis towards immune-protective and pro-tumorigenic macrophage populations. However, the latter results were not observed in a second in vivo experiment, in which the sizes of the metastases were larger (Appendix A).

T cells play an important role in the outgrowth of metastasis, where cytotoxic T cell responses need to be evaded or repressed for successful metastatic outgrowth [23,24]. Staining for T cell markers CD3 and CD8 revealed a reduction of both CD3^+^ (Figure 5E and Appendix A) and CD3^+^/CD8^+^ T cells (Figure 5F and Appendix A) in the caALK5 expressing liver metastasis compared to the control. This was corroborated in a second mouse experiment, in which the metastases were larger (Appendix A). Overall, the presence of CD3^+^ and CD3^+^/CD8^+^ T cells in the liver metastases was relatively high, despite B16F10 being a poorly immunogenic tumor cell line.

### 2.6. Secretome Analysis Suggests Tumor Microenvironmental Changes upon caALK5 Expression

The characterization of the B16F10 tumor microenvironment revealed changes in mostly immune cell compartments. To investigate if caALK5 expression in B16F10 cells could contribute to these changes, we examined the secretome of these B16F10 cells upon caALK5 expression in vitro. A mass spectrometry analysis was performed on newly synthesized proteins in the conditioned medium of B16F10 caALK5 cells under control conditions or dox treatment for 48 h. Amongst the most highly upregulated proteins in the caALK5 expressing condition, we found known TGF-β target proteins PAI-1 (*Serpine1),* CTGF (*Cnn2),* and Periostin (*Postn)* (Figure 6A). PAI-1 and CTGF have roles in matrix remodeling, while Periostin has been shown to recruit BMDM [25,26,27,28]. With this data, a protein signature was created for caALK5 expressing B16F10 cells (Appendix A, gene names corresponding to the encoded proteins are shown). Analysis of enrichment of pathway and biological processes showed no direct evidence for immune suppressive processes upon caALK5 expression (Figure 6B), while reduced number of T cells were found in vivo. This suggests a more indirect effect of B16F10 cells on immune cells. Gene set enrichment did show an enrichment in various extracellular matrix related processes (Figure 6B), indicating that matrix could be remodeled in caALK5 B16F10 liver metastasis. This could lead to a physical barrier and limit the penetration of T cells. Indeed, we observed a reduced number of T cells within the caALK5-expressing B16F10 liver metastases (Figure 6C). However, future studies are needed to investigate the underlying mechanisms in more depth.

## 3. Discussion

Despite the developments in targeted therapies and immunotherapies, the prognosis for metastatic melanoma patients remains unfavorable [29]. Especially melanoma patients presenting with liver metastasis have a poor prognosis [1,5]. Therefore, it is important to understand the development of liver metastasis. The multifunctional cytokine TGF-β has been implicated in melanoma progression and invasion [9,10,11,12]. Additionally, TGF-β can influence pro-metastatic processes mediated by various cell types in the liver [1,7]. Here, we investigated the role of the multifunctional cytokine TGF-β in the process of liver metastasis formation in B16F10 tumor cells and the tumor microenvironment. We engineered B16F10 cells to have doxycycline inducible expression of a constitutively active or kinase-inactive TGF-β type I receptor, ALK5, leading to TGF-β pathway activation or repression respectively. In vitro, TGF-β activation repressed B16F10 cell proliferation and migration. In contrast, ectopic expression of caALK5 in vivo increased the metastatic liver outgrowth of B16F10 cells in the liver. Interestingly, inhibiting TGF-β in the tumor microenvironment did not affect metastatic liver outgrowth. When characterizing the tumor microenvironment of B16F10 liver metastasis, we discovered a change towards an immune suppressive tumor microenvironment upon caALK5 expression. To investigate what changes in the tumor microenvironment could be due to caALK5 expression in B16F10 cells, the secretome of caALK5 expressing and control B16F10 cells were compared. This resulted in a protein signature including TGF-β targets Serpine 1, CTGF and Periostin, and other matrix remodeling related proteins. The secretome analysis hints towards microenvironmental changes resulting in fewer T cells upon caALK5 expression that could enhance outgrowth in vivo. Future studies and validation in other melanoma metastasis models will have to confirm whether matrix remodeling is underlying the observed tumor microenvironmental changes and enhanced tumor outgrowth in vivo.

The liver is one of the main metastatic sites of melanoma, but metastases are also frequently found in other organs including bone, brain and lungs. To investigate whether the presented results are liver specific, melanoma mouse models in immune competent mice, presenting with metastases in other organs, should be investigated. Cell type as well as organ specific effects of TGF-β and TGF-β inhibition are likely. Interestingly, immunotherapy was found to be less effective for liver metastasis, as shown by Yu et al. [5]. This would make the presented liver metastasis model an attractive melanoma metastasis model for optimizing therapy response.

The presented results hint that TGF-β-signaling B16F10 cells remodel their tumor microenvironment in the liver, leading to a reduced number of T cells. This would be in line with previous findings on TGF-β and the immune tumor microenvironment [24,30,31]. The results reported here are most likely not the result of direct inhibition of the immune cells by local TGF-β, or other cytokines by caALK5 expressing B16F10 cells, as TGF-β inhibition did not result in changes in tumor outgrowth and no immune suppressive cytokines were found to be enriched in the secretome analysis. However, the three proteins that were found to be highly upregulated upon caALK5 expression, PAI-1, CTGF, and Periostin, have been associated with melanoma prognosis and/or poor immunotherapy response [32]. PAI-1 has been reported to regulate PD-L1 expression in melanoma cells, as well as promote pro-inflammatory M2 macrophages, mediating immune evasion and the reduced efficacy of anti-PD-L1 therapy in melanoma patients [33,34]. Periostin is a potent (CCR2^+^) BMDM recruiter, as evident from studies on glioblastoma and primary melanoma lesions [25,26,27,28]. Results from Yu et al.’s study show liver-specific results of CD8^+^ T cell killing mediated by recruited BMDM, resulting in a poor response to immunotherapy [5]. As we observed a potential upregulation of CCR2^+^ BMDM and a reduction in CD8^+^ T cells in our caALK5 B16F10 liver metastasis, it will be interesting to investigate whether TGF-β could play an underlying role in immunotherapy resistance, either directly or indirectly through resident or bone marrow-derived macrophages. We observed a reduction in infiltrated T cells upon TGF-β signaling in vivo. This could indicate an interplay with B16F10 cells and surrounding fibroblasts, leading to fibroblast activation and the formation of a protective border, as seen in colorectal tumor [35]. In our data, PDGFR-β staining, indicative of CAFs, did not show any changes; however, secretome analysis showed an upregulation of the extracellular matrix and remodeling proteins upon TGF-β signaling. The latter warrants further investigation of the involvement of extracellular matrix and CAFs inB16F10 liver metastasis formation.

The immune-suppressive effects of TGF-β signaling have gained a lot of interest with regards to anti-tumor therapies [5,15,30]. TGF-β has been linked to direct and indirect immune evasion in colorectal and other tumor types [24,35], and combinations of TGF-β inhibitors with immune checkpoint blockade can elicit synergistic therapeutic effects [24,30]. This has encouraged therapeutic exploration of the combination of TGF-β inhibitors and immunotherapies to relieve tumor-induced immune suppression [36,37]. In the model used in this study, blocking TGF-β signaling in tumor cells will most likely be of more interest than blocking microenvironmental TGF-β. While our study did not show any effect of B16F10 metastatic liver outgrowth upon 1D11 treatment, it will be of interest to repeat 1D11 treatment or TGF-β inhibition in combination with immunotherapy treatment.

The presented study focusses on TGF-β induced changes in the tumor microenvironment, but does not investigate TGF-β response in B16F10 cells themselves. However, a rewiring of the B16F10 response to TGF-β signaling may have also occurred in vivo in order to overcome the negative effects on cell proliferation. Dedifferentiation of melanoma cells, resulting in depigmentation and migration of the tumor cells, is known to be promoted by TGF-β in vitro and in vivo [38]. Crosstalk with other pathways can contribute to the rewiring to TGF-β stimulation as well [39]. Additionally, behavior of B16 cells upon TGF-β treatment could be matrix dependent, as an increase of migration was observed in vitro in different collagen-based matrixes [12,38]. Intravital imaging of B16F10 cells can give more insights in their behavior during liver metastasis.

## 4. Methods

### 4.1. Reagents

Human TGF-β3 (a kind gift from A. Hinck, University of Pittsburgh, Pittsburgh, PA, USA) was dissolved in 4 mM HCl/0.1% recombinant bovine serum albumin (BSA). TGF-β3 (1 ng/mL), doxycycline (D9891 Sigma, Darmstadt, Germany, 1 μg/mL), selective small molecule ALK5 kinase inhibitor SB505124 (1 μM, #3263, Tocris, Abingdon, UK), and 1D11, a pan-TGF-β neutralizing antibody (#MAB1835, 10 μg/mL (R&D Systems, Abingdon, UK), were used in the cell culture experiments.

### 4.2. Generation of Plasmids

Wildtype ALK5 (wtALK5)-flag and ALK5^T204D^ (constitutively active (ca)ALK5)-flag cDNA fragments were generated through PCR using an ALK5 FW primer containing a Bmtl restriction digestion site and an ALK5-flag REV primer. These fragments were inserted in the pLIX401 backbone using Age1 and Bmtl1 digestion, resulting in pLIX401-wt/caALK5. From here, the wt/caALK5 flag fragments were amplified using ALK5 FW and ALK5 REV PCR primers, followed by digestion with Sal1 and Xho1 to insert the DNA fragments into the p.ENTR 1a vector. The resulting p.ENTR 1a-wtALK5 vector was used to create a kinase inactive (ki) ALk5 construct p.ENTR 1a-ALK5^K232R^ through mutagenesis. p.ENTR 1a ki/caALK5 constructs were then used to generate p.Inducer20-neo-wt/ca ALK5 constructs using gateway cloning. P.Inducer20 was obtained from Stephen Elledge (Addgene plasmid# 44012; RRID:Addgene_44012) [17].The final plasmids were verified using control restriction enzyme digestion and DNA sequencing. The primers used for cloning can be found in Appendix A.

### 4.3. Cell Culture

B16F10 cells (CRL-6475, ATCC) were cultured in Dulbecco’s modified eagle medium (DMEM) containing 10% fetal bovine serum (FBS) and 100 U/mL penicillin-streptomycin (15140122; Gibco, Bleiswijk, The Netherlands). CM cells were a kind gift from Prof. Helfrich (Essen, Germany) [40] and were cultured in Roswell Park Memorial Institute (RPMI) 1640 containing 10% FBS and 100 U/mL penicillin–streptomycin. Cells were cultured in 37 °C/5% CO_2_ incubators and were regularly tested for the absence of mycoplasma infections.

### 4.4. Western Blot Analysis

Protein samples from cells were generated through lysis with Leammli buffer (0.12 M Tris-HCl pH 6.8, 4% SDS, 20% glycerol, 35 mM β-mercaptoethanol and bromophenol blue). Protein samples from flash-frozen liver and liver metastasis tissue were made in a radioimmunoprecipitation assay (RIPA) buffer. The liver samples were not corrected for contribution of liver metastasis per sample. Protein concentrations were measured using detergent-compatible (DC) protein assay (#5000111, Bio-Rad, Hercules, CA, USA).). In total, 25–30 μg of protein was loaded onto 10% sodium dodecyl sulfate (SDS)-polyacrylamide gels. Western blots were performed using previously published protocols using 5% non-fat dry milk powder as the blocking solution. The following primary antibodies in 2.5% non-fat dry milk powder were used: ALK5 for in vitro protein samples (V22, SC-398, Santa Cruz, CA, USA), ALK5 for in vivo protein samples (ab235578, Abcam, Cambridge, UK) plasminogen activator inhibitor (PAI)-1 (ab222754, Abcam, Cambridge, UK), phospho-SMAD2 [41], Flag (M2, F1804 Sigma, Darmstadt, Germany), Vinculin (V9131, #MA5-11690 Sigma, Darmstadt, Germany), and GAPDH (6C5, #MAB374, Sigma). Chemiluminescence images were taken using a Chemidoc (Bio-Rad, Hercules, CA, USA).

### 4.5. RT-qPCR

Total RNA extraction was performed using the NucleoSpin RNA II kit (Macherey-Nagel, Dueren, Germany) according to the manufacturer’s instructions. cDNA was generated using the Revert Aid First-Strand cDNA synthesis kit (Thermo Fisher, Bleiswijk, The Netherlands). Quantitative PCR was performed using SYBBR GoTaq qPCR master mix (Promega, Leiden, Netherlands) and 0.5 μM of primers. RT-qPCR was performed on the CFX connect Real-Time PCR detection system (Bio-Rad, Hercules, CA, USA). The primers used are listed in Appendix A. Experiments are performed in technical duplicates and biological triplicates, target gene expression was normalized to the geometric mean (geomean) of *Gapdh* and *Hprt* expression.

### 4.6. Animal Studies

All animal experimental protocols were approved by the animal welfare committee of the Leiden University Medical Center (LUMC) and the Dutch Animal Experiments Committee. Experiments were performed under license number VD116002016705.

Eight-week-old female C57BL/6J mice were used for the animal experiments. A mouse and rat antibody production (MAP/RAP) test was performed on B16F10 cells prior to the animal experiments. B16F10 cells containing caALK5 or kiALK5 constructs were treated with control or doxycycline-containing medium (1 μg/mL) one day prior to injection. Mice were put on control or doxycycline-containing chow (625 mg/kg, #A153D70620, Bio-Services, Uden, The Netherlands) one day prior to surgery. Mice were injected with 5 × 10^5^ B16F10 cells containing caALK5 or kiALK5 expression constructs using a mesenteric vein injection [18]. Doxycycline-containing food was replaced every two to three days. In the indicated experiments, 1D11 (5 mg/kg, clone 1D11.16.8, #BP0057 BioXcell, Lebanon, NH, USA) or isotype control mouse IgG1 (5 mg/kg, clone MPC-21, #BP0083) was administered using intraperitoneal injection three times per week. Mice were weighed and monitored for animal well care 3 times per week and sacrificed after 14 days using CO_2_ inhalation.

Livers were harvested and photographed before fixation in PLP buffer (1% paraformaldehyde, 0.2% NaIO_4_, 61 mM Na_2_HPO_4_, 75 mM L-lysine and 14 mM NaH_2_PO_4_ in H_2_O) overnight at 4 °C. After fixation, organs were incubated in sucrose for 6–12 h at 4 °C. Livers were embedded in optimal cutting temperature compound (OCT) and cryosectioned into 10 μm sections using a Leica cryostat at four different depths in the liver, 500 μm apart.

### 4.7. Immunofluorescence Staining

Liver sections were rehydrated with phosphate-buffered saline (PBS) for 10 min and permeabilized with 0.1% Triton for 5 min. Tissues were blocked in blocking buffer (PBS containing 2% BSA and 5% normal goat serum) for 30 min. Antibodies were diluted in 0.5× blocking buffer. Primary antibodies were incubated overnight at 4 °C. Secondary antibodies were incubated for one hour at room temperature. Nuclei were stained using vectashield mounting medium containing 4′,6-diamidino-2-phenylindole (DAPI) (H-1500, Vector Laboratories, Newark, CA, USA). Slides were scanned using the Axio Scan.Z1 slide scanner (Zeiss, Munster, Germany) and analyzed with QuPath software [42]. The fluorescent signal was analyzed using semi-automatic scripts in QuPath, and conditions were blinded during analysis. The following primary antibodies were used: platelet-derived growth factor receptor (PDGFR)-β (1/100, #31695, Cell Signaling Technology, Leiden, The Netherlands), CD3 (1/100, #NB600–1441SS, Novus Bio, Abingdon, UK), CD8 (1/100, #14-0808-82, Invitrogen, Bleiswijk, The Netherlands), plasminogen activator inhibitor (PAI)-1 (1/ab222754, Abcam), ionized calcium-binding adapter molecule1 (Iba-1) (1/1000, #019-19741, Fujifilm, Dusseldorf, The Netherlands), C-C chemokine receptor type 2 (CCR2) (1/100, ab273050, Abcam, Amsterdam, The Netherlands), and C-type lectin domain family 4 (CLEC4F) (PA5-47396, Thermofisher, Bleiswijk, The Nethelands). The following secondary antibodies were used: goat anti-mouse 594, goat anti-rat 594, goat anti-rabbit 594, goat anti-rabbit 647, and goat anti-rat 488 (all 1/250, Thermofisher, Bleiswijk, The Netherlands).

To calculate the hepatic replacement area, metastases were detected based on H2B-mTurquoise staining of the tumor cells. The total area of metastasis was quantified and set as a percentage to the total area of liver analyzed.

### 4.8. Fluorescence-Activated Single Cell Sorting (FACS)

Prior to FACS sorting, cells were collected in PBS containing 0.2% BSA and filtered through a 70 μm filter. The cells were sorted for high expression of histone 2B (H2B)-mTurquoise using the Aria I 4L (BD Biosciences, San Jose, CA, USA) at the Flow cytometry Core Facility (FCF) of LUMC in Leiden, Netherlands. Samples were sorted at 4 °C. For clonal cell line generation of B16F10 caALK5 and kiALK5 cells, cells were sorted in a 96-well plate containing cell culture medium.

### 4.9. Incucyte^®^ Proliferation Assay

B16F10, B16F10 caALK5, and B16F10 kiALK5 cells were seeded in a 96-well plate and treated with TGF-β3 (1 ng/mL), SB505124 (1 μM) or doxycycline (1 μg/mL) for two days. Cells were imaged every two hours on the Incucyte system (Goettingen, Germany), and confluency was measured using Incucyte^®^ software.

### 4.10. MTS Assay

B16F10, B16F10 caALK5, and B16F10 kiALK5 cells were seeded in a 96-well plate and treated with TGF-β3 (1 ng/mL), SB505124 (1 μM) or doxycycline (1 μg/mL) for three days. MTS reagent, used to measure metabolic activity, was added after one, two, and three days, and absorbance was measured at 490 nm.

### 4.11. Colony Formation Assay

In total, 500 B16F10 cells were plated and treated with vehicle control (DMSO), TGF-β3 (1 ng/mL), and/or SB505124 (1 μM). The medium was refreshed every 2–3 days. After 7 days, cells were washed with PBS and stained with crystal violet (1%) for 20 min.

### 4.12. Wound Healing and Invasion Assay

In total, 2 × 10^4^ B16F10 cells were plated in a 96-well plate to achieve semi-confluency. Cells were treated with control or doxycycline (1 μg/mL) for 3 days and serum starved in 0.5% FBS for 16 h before creating a wound using the 96-well woundmaker^®^ (#4563, Essen bioscience, Newark, UK). During wound healing, cells were treated with TGF-β3 (1 ng/mL), SB505124 (1 μM) or doxycycline (1 μg/mL) in cell culture medium containing 0.5% FBS. Cells were imaged every 2 h on the Incucyte^®^ system, and wound width was calculated using Incucyte^®^ software. For the wound invasion assay, the wound was filled with 2.5 mg/mL Matrigel (growth factor reduced, Corning, Amsterdam, The Netherlands) and the experiment was performed in 0.1% FBS. Wound density was calculated using Incucyte^®^ software.

### 4.13. Random Migration Assay

The B16F10 cells were sparsely plated. The next day, the cells were washed in PBS, and DMEM containing 0.1% FBS was added. The cells were treated with the control or TGF-β3 (1 ng/mL). The cells were then imaged using a live cell AF6000 CRTD microscope (Leica, Amsterdam, The Netherlands). Pictures were taken every 30 min at 3 positions per condition, measuring 10 cells per position. Migration and displacement were calculated using ImageJ; the migration speed is shown as μm/min.

### 4.14. Mass Spectrometry of B16F10 Secretome

B16F10 cells were cultured in four 100 mm cell culture dishes on cell culture medium consisting of DMEM/F12 (Gibco, Bleiswijk, The Netherlands) medium supplemented with 10% FBS (Gibco), 2 mM glutamine (Gibco), 100 U/mL penicillin and 100 μg/mL streptomycin (Gibco) at 37 °C, 5% CO_2_. Two dishes were treated with 1 µg/mL doxycycline for 48 h prior to medium collection. After 24 h, cell culture medium was aspirated and cells were incubated with PBS for 30 min at 37 °C to deplete intracellular methionine. Next, the cells were cultured for 24 h in cell culture medium prepared with DMEM/F12 medium without methionine (Gibco), supplemented with 0.1 mM AHA (L-Azidohomoalanine) (ThermoScientific, Bleiswijk, The Netherlands) to label nascent proteins, and 1 µg/mL doxycycline where appropriate. AHA-labelled conditioned medium was collected and concentrated to 250 μL using 3 kDa centrifugal filters (Amicon, Darmstadt, Germany) and 1×complete protease inhibitors (CPIs, Merck, Amsterdam, The Netherlands were added. Samples were snap frozen and stored at −80 °C until further processing. Enrichment of AHA-labeled proteins and on-bead digestion was performed as described in [43]. The digest was then desalted and concentrated on C18 StageTips without acidification [44]. Peptide labeling was done by dimethyl labeling [45], and StageTips were stored at 4 °C until measurement by LC-MS/MS. Secretomics mass spectrometry and data analysis were performed as described in [43] and protein identification and quantification was done in MaxQuant v1.5.7.1 [46] with standard settings and requantify enabled. Common contaminants and decoy database hits were removed from the resulting MaxQuant proteinGroups file and alias gene names were replaced with official gene symbols using the Limma package [47]. In case of duplicate entries, the entry with the highest number of Andromeda score was retained. Protein groups were required to have at least two assigned peptides, of which at least one was a unique peptide, and only protein groups with a quantified ratio for the forward and reverse reaction were considered for further analysis. Statistical significance thresholds were set at Q3 + 1.5 × IQR and Q1 − 1.5 × IQR, and proteins were required to exceed these thresholds in both the forward and reverse run to be labeled as significant.

### 4.15. Gene Set Enrichment Analysis

Protein names from the protein signature identified with the secretome analysis were converted to gene names. The resulting signature was analyzed for enrichment in biological processes and signaling pathways using Metascape (http://metascape.org, accessed on 4 February 2023). Gene ontology terms, Kyoto Encyclopedia of Genes and Genomes (KEGG) pathways and Reactome Gene Sets were used by Metascape [48,49].

### 4.16. Statistical Analysis

Experiments were performed in independent biological triplicates, unless otherwise indicated. Error bars represent ± standard error of the mean (SEM), unless otherwise indicated. Statistics were calculated using GraphPad Prism 7 software, and statistical significance was defined as ns (*p* ≥ 0.05), * *p* ≤ 0.05, ** *p* ≤ 0.01, *** *p* ≤ 0.001 and **** *p* ≤ 0.0001. Post-hoc analysis was performed if a significant interaction was found.

## 5. Conclusions

We present a dox inducible model to activate TGF-β signaling in vitro and in vivo in mouse melanoma B16F10 cells. We demonstrate that the activation of ALK5 signaling in B16F10 cells in vitro inhibits cell proliferation and migration, while the in vivo activation of ALK5 in B16F10 cells can increase the metastatic liver outgrowth in the liver. Matrix remodeling and changes in immune cell presence induced by caALK5 expression, or rewiring of the TGF-β response in B16F10 cells could be underlying these changes, although the exact mechanism still remains to be elucidated.

## Figures and Tables

**Figure 1 ijms-24-08676-f001:**
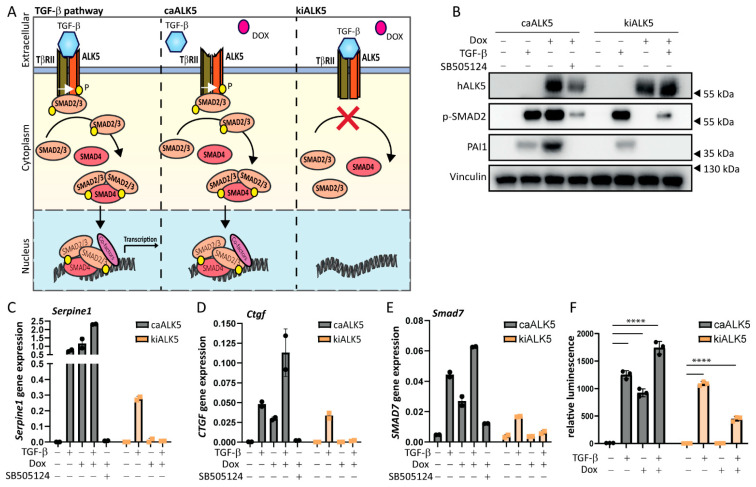
Doxycycline-inducible model to activate or repress TGF-β type I receptor signaling. (**A**) Schematic overview of the doxycycline (dox, indicated with a red dot)-inducible model expressing caALK5 or kiALK5 to activate or repress TGF-β signaling. (**B**) Western blot analysis for the expression levels of human ALK5 (hALK5), phosphorylated SMAD2, and PAI-1 in cell lysates from B16F10 caALK5 and kiALK5 cells that were treated with dox (1 μg/mL) and/or TGF-β (1 ng/mL). Vinculin was used as a loading control. Representative of 3 biological replicates. (**C**–**E**) qPCR analysis of *Serpine1* (**C**), *Ctgf* (**D**), and *Smad7* (**E**) in B16F10 caALK5 and kiALK5 cells treated with doxycycline (1 μg/mL) and/or TGF-β (1 ng/mL) and/or SB505124 (1 μM). Expression was normalized to the housekeeping genes *Hprt* and *Gapdh*. Representative of 3 biological replicates, 2 technical duplicates shown, mean ± S.D. (**F**) CAGA_12_-luciferase reporter assay of B16F10 cells treated with doxycycline (1 μg/mL) and or TGF-β (1 ng/mL). Three biological replicates. Bars on the graph indicate the mean ± S.E.M. **** *p ≤* 0.0001, two-way ANOVA.

**Figure 2 ijms-24-08676-f002:**
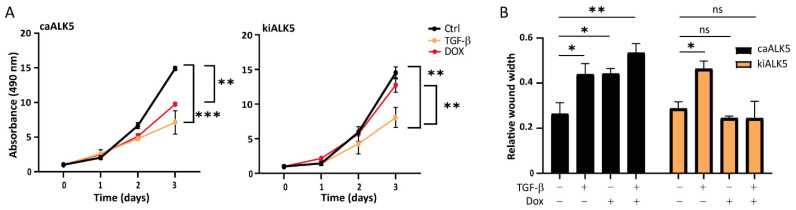
TGF-β signaling reduces cell proliferation and the migration of B16F10 cells. (**A**) MTS assay of B16F10 caALK5 and kiALK5 cells treated with TGF-β (1 ng/mL) and/or dox (1 μg/mL) for indicated time points. The assay was performed in 10% serum. n = 3 biological replicates, mean ± S.E.M. ** *p* ≤ 0.01, *** *p* < 0.001, one way ANOVA. (**B**) Wound healing assay was performed using the Incucyte^®^ system under serum-starved conditions. Prior to wounding, cells were pre-treated with the control or doxycycline (1 μg/mL) to induce the expression of caALK5 or kiALK5. During the scratch assay, cells were treated with TGF-β3 (1 ng/mL) and dox (1 μg/mL). The wounded area was followed by live cell microscopy; the relative wound width after 16 h is shown. n = 3 biological replicates, mean ± S.E.M. ns *p* ≥ 0.05, * *p* ≤ 0.05, ** *p* ≤ 0.01 one way ANOVA.

**Figure 3 ijms-24-08676-f003:**
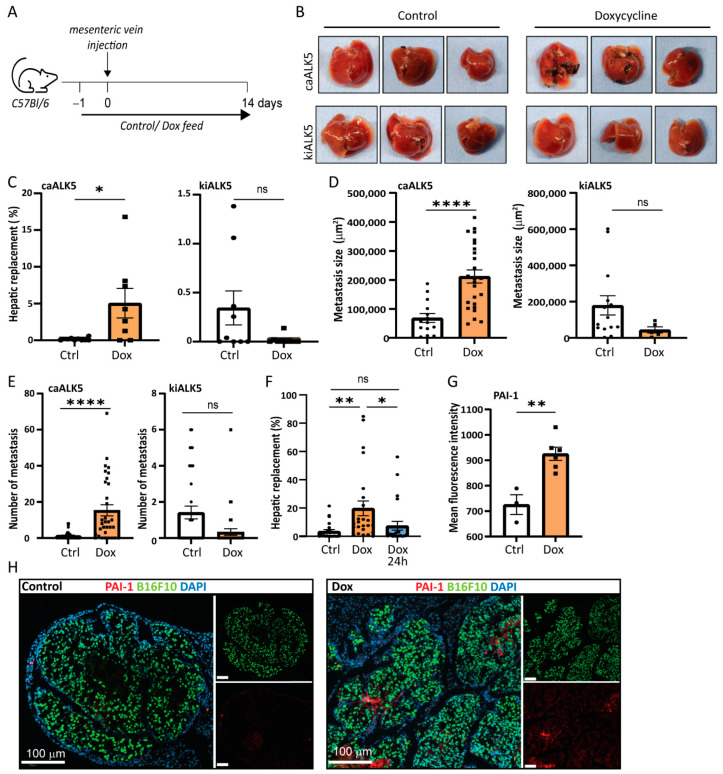
Increased TGF-β type I receptor signaling increases metastatic outgrowth in liver of B16F10 cells. (**A**) Schematic outline of the experimental set-up of the in vivo experiment, injecting B16F10 caALK5 or B16F10 kiALk5 cells, 9 mice per condition. Mice were fed control chow or chow containing dox (625 mg/kg). (**B**) Examples of metastatic liver outgrowth after injection of B16F10 caALK5 and B16F10 kiALK5 cells in mice under control chow or dox-containing chow (625 mg/kg). (**C**) Metastatic replacement in liver of mice after the injection of B16F10 caALK5 and B16F10 kiALK5 cells in the control and dox-treated group. caALK5 expression increased the metastatic liver outgrowth of B16F10 cells (control n = 9 mice, dox n = 8), while kiALK5 expression reduced metastatic liver outgrowth (control n = 9, dox n = 9). Average metastatic liver replacements in 4 different sections per mouse liver are shown. Mean ± S.E.M. Student’s *t*-test * *p* ≤ 0.05, ns *p* ≥ 0.05. (**D**) Size (in μm^2^) of all liver metastasis of B16F10 caALK5 (control n = 14, dox n = 25) and kiALK5 (control n = 14, dox n = 5). Data from 4 sections per mouse liver are shown. Student’s *t*-test, **** *p* ≤ 0.0001. (**E**) Number of all liver metastasis of B16F10 caALK5 (control n = 36, dox n = 32) and kiALK5 (control n = 36, dox n = 36). Data from 4 sections per mouse liver are shown. Mean ± S.E.M, Student’s *t*-test, ns *p* ≥ 0.05, **** *p* ≤ 0.0001. (**F**) Metastatic liver replacement of B16F10 caALK5 liver metastasis treated with control or doxycycline-containing chow for two weeks (dox) or 24 h (dox 24 h). Data from 4 sections per mouse liver are shown. n = 24, mean ± S.E.M, ** *p* ≤ 0.01, * *p* ≤ 0.05 one-way ANOVA. (**G**) Analysis of PAI-1 protein expression by immunohistochemical staining in liver metastasis of mice injected with B16F10 caALK5 cells. Control n = 3, dox n = 6, average of metastases per section. Mean fluorescence intensity was measured within metastasis using Qupath, mean ± S.E.M ** *p* ≤ 0.01, Student’s *t*-test. (**H**) Examples of (**G**).

**Figure 4 ijms-24-08676-f004:**
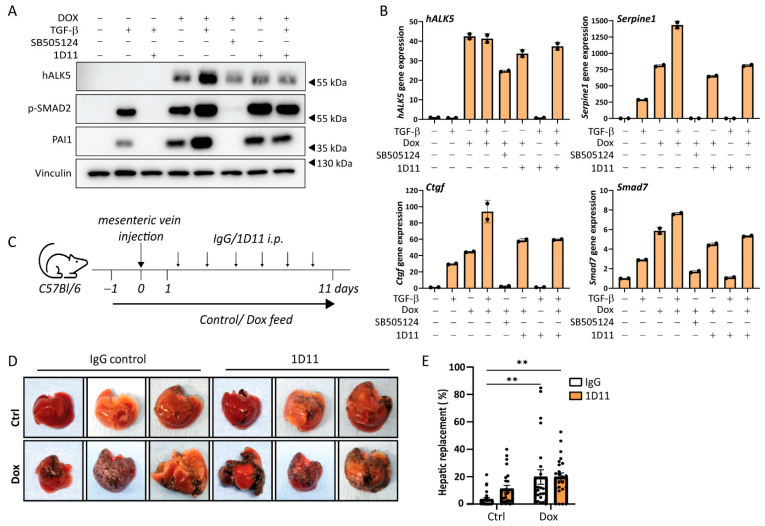
TGF-β neutralizing antibody does not affect caALK5-mediated effects on the metastatic outgrowth in liver of B16F10 cells. (**A**) Analysis of the protein expression of TGF-β signaling components by Western blot of B16F10 caALK5 cells treated with dox, TGF-β3, SB505124 or TGF-β-neutralizing antibody 1D11. Vinculin level was used as a loading control. (**B**) Analysis of gene expression by qPCR in B16F10 caALK5 cells treated without or with dox (1 μg/mL), TGF-β3 (1 ng/mL), SB505124 (1 μM) or TGF-β-neutralizing antibody 1D11 treatment (10 μg/mL). Effects of caALK5, TGF-β and 1D11 treatment is measured by mRNA levels of *Serpine1*, *Ctgf*, and *Smad7*. Gene expression was corrected by the housekeeping genes *Gapdh* and *Hprt*. Mean ± S.D., technical duplicates. Representative of 3 biological repeats. (**C**) Schematic representation of the experimental set-up of in vivo experiment. Mice were injected with B16F10 or B16F10 caALK5 cells and were treated with control chow or dox-containing chow (625 mg/kg) for two weeks. Mice were injected with IgG control or 1D11 antibody (5 mg/kg), 6 mice per condition. (**D**) Examples of livers containing B16F10 or caALK5 expressing B16F10 liver metastasis, treated with IgG or 1D11. (**E**) Metastatic liver replacement was calculated for 4 sections per liver of mice from in vivo experiment shown in (**C**,**D**). n = 24, mean ± S.E.M. ** *p* ≤ 0.01, two-way ANOVA.

**Figure 5 ijms-24-08676-f005:**
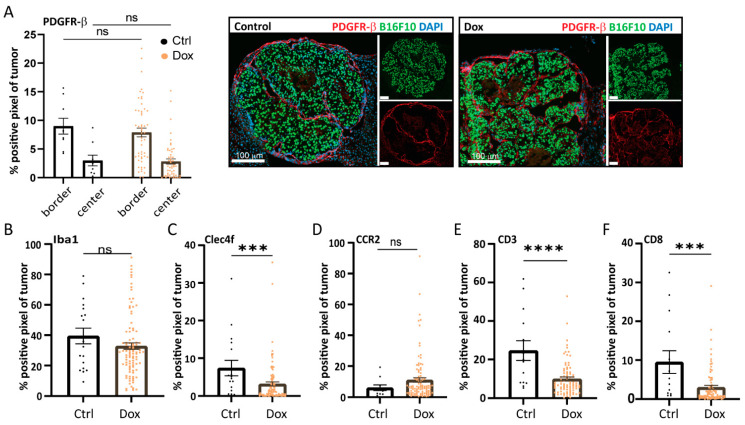
Increased TGF-β receptor-I signaling in B16F10 alters the tumor microenvironment in liver metastases. Analysis of the immunohistochemical staining of microenvironmental cell markers in B16F10 caALK5 liver metastases treated with and without dox. All metastases per group are shown, from n = 9–14 in the control and n = 84–101 in the dox-treated group. The percentage of positive staining pixels in metastases was calculated; analysis was performed using QuPath. Mean ± S.E.M. Significance was calculated using Student’s *t*-test, ns, *** *p* ≤ 0.001, **** *p* ≤ 0.0001. Staining was performed for the CAF marker PDGFR-β (**A**), general macrophage marker Iba1 (**B**), liver-resident macrophage marker Clec4f (**C**), bone marrow-derived macrophage marker CCR2 (**D**), general T cell marker CD3 (**E**), and cytotoxic T cell marker CD8 (**F**).

**Figure 6 ijms-24-08676-f006:**
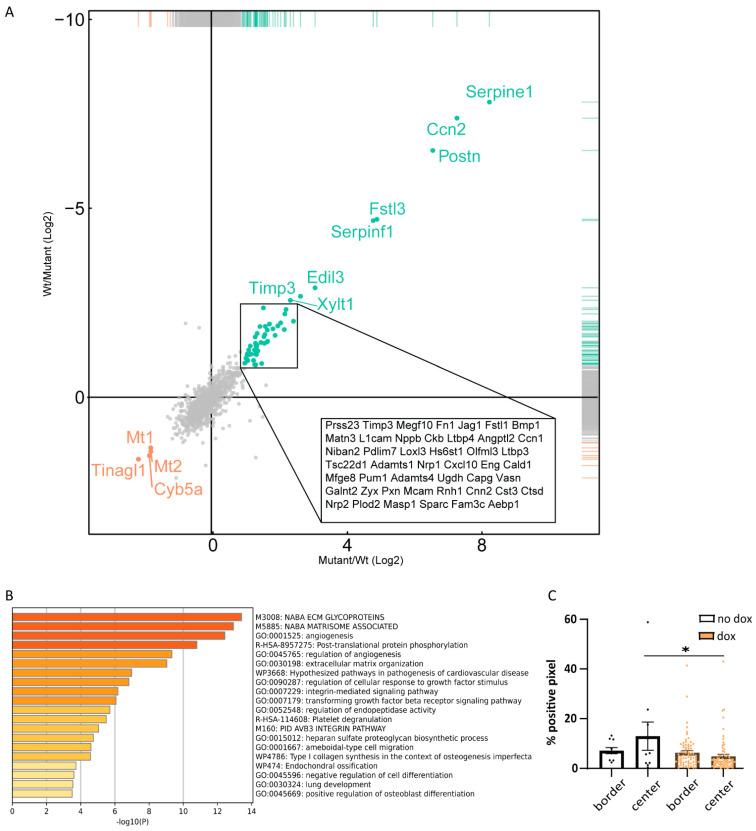
Secretome analysis of B16F10 caALK5 cells. (**A**) Mass spectrometry analysis of the secretome of control and caALK5 expressing cells. Cells were labeled with azidohomoalanine and treated with control or dox (1 μg/mL) for 48 h prior to biotin labeling and pulldown, followed by mass spectrometry analysis. Significantly upregulated (green) and downregulated (red) proteins in caALK5 expressing B16F10 secretome are shown. (**B**) Proteins upregulated in caALK5 expressing B16F10 cells were analyzed for enrichment in GO annotation and KEGG pathways using Metascape. Top 20 significantly upregulated processes are shown, colored by *p* values. (**C**) Analysis of CD3 staining in the tumor border and center. All metastases per group are shown, from n = 10 in the control and n = 70 in the dox-treated group. Significance was calculated using two-way ANOVA, * *p* ≤ 0.05.

## Data Availability

The data presented in this study are available in Appendix A.

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
