# Peer review of "TGF-β Type I Receptor Signaling in Melanoma Liver Metastases Increases Metastatic Outgrowth"

_ijms, 2023, doi:10.3390/ijms24108676_

Round 1
Reviewer 1 Report
By using dox-inducible constitutively-active and kinase-dead versions of the type I receptor for TGF-b, the authors show that activation of the type I receptor for TGF-b promotes metastatic growth in the liver of B16F10 melanoma cells. The study is straightforward and the conclusions supported by the data shown. This paper will be suitable for publication after some revision.
Specific points
1. I think that the expression “hepatic outgrowth” is misleading. One gets the impression that it is the hepatocytes and other normal cells in the liver that grow. A better expression would be “metastatic growth”.
2. The authors focus on metastases. It would be interesting to also know the effect of TGF-b signaling on the growth of the primary tumor.
3. Fig. 1B: Lanes and text should be better aligned.
4. Fig. 1C-D: No treatment for kiALK5 are missing. Moreover, bars 2 and 3 from the right have probably been mixed up (also the case for Fig. 1F).
5. Line 368: The reference should be to Fig. 3B, not 1B.
6. Line 487: The text in the legend refers to panel G, which is not there.
7. A number of typos and other small mistakes should be corrected. Examples include line 92 (10 mg/mL), line 132 (RT-qPCR); line 185 (FACS); line 222 (“Mass spectrometry of B16F10 secretome” should be headline 2.14 and in italics, also then shifting the numbering of the following sections); lines 279 and 285-286 (there is another type face for some of the text on these lines); lines 322 and 435-436 (genes should be in italics); line 350 (“time periods”); references to P-values should be consistent.
Author Response
We thank the reviewer for reading our manuscript thoroughly and providing us with insightful and useful comments. The paper in its revised form has improved greatly because of it. Below we adress the reviewers concerns one by one.
- I think that the expression “hepatic outgrowth” is misleading. One gets the impression that it is the hepatocytes and other normal cells in the liver that grow. A better expression would be “metastatic growth”.
We agree with the reviewer that this might be confusing to the reader. However, we did like to emphasize that the metastases grow in the liver; as such we changed “hepatic outgrowth” to “metastatic liver outgrowth” throughout the manuscript. We have had our manuscript proofread by MDPI (see certificate in uploaded files). In addition, we have carefully gone through the manuscript again and corrected spelling mistakes.
- The authors focus on metastases. It would be interesting to also know the effect of TGF-b signaling on the growth of the primary tumor.
We agree that investigating the role of TGF-b in primary melanoma is also of interest. However, in our paper, we focused on the role of TGF-b in controlling melanoma metastasis. Patients presenting with primary melanoma without metastasis have a good prognosis, compared to the very poor prognosis of metastatic melanoma patients. Therefore, understanding molecular mechanisms that underlie metastasis formation could results in better treatment strategies for metastatic melanoma patients. To investigate the role of TGF-b in primary melanoma (which is likely also influenced by TME) is beyond the scope of the current paper.
- Fig. 1B: Lanes and text should be better aligned.
We have now better aligned the text with the lanes.
- Fig. 1C-D: No treatment for kiALK5 are missing. Moreover, bars 2 and 3 from the right have probably been mixed up (also the case for Fig. 1F).
We apologize for this mistake and thank the reviewer for noticing this. We have replaced this panel with a new figure including the no treatment for kiALK5 condition, and we have adjusted the order in Figure 1B and 1F.
- Line 368: The reference should be to Fig. 3B, not 1B.
Thank you, we have changed the reference to Fig. 3B
- Line 487: The text in the legend refers to panel G, which is not there.
We have adjusted the Figure legend; we thank the reviewer for pointing this out.
- A number of typos and other small mistakes should be corrected. Examples include line 92 (10 mg/mL), line 132 (RT-qPCR); line 185 (FACS); line 222 (“Mass spectrometry of B16F10 secretome” should be headline 2.14 and in italics, also then shifting the numbering of the following sections); lines 279 and 285-286 (there is another type face for some of the text on these lines); lines 322 and 435-436 (genes should be in italics); line 350 (“time periods”); references to P-values should be consistent.
We thank the reviewer for pointing out these small mistakes. We have now corrected them in the revised manuscript. Additionally, we have attached a certificate that a spell check has been performed by MDPI for this paper. We also have carefully gone through the paper and corrected any additional spelling errors.
We attached the edited revised manuscript and hope that the revised manuscript satisfies all the concerns of the reviewer.
Reviewer 2 Report
The manuscript “TGF-b type I receptor signaling in melanoma liver metastases increases hepatic outgrowth” by Marvin and colleagues describes the effect of TGF-beta or its downstream signalling on the in vitro and in vivo growth of B16-F10 melanoma cells.
The paper is carefully conducted in most parts and contains interesting information regarding the biological output of this signaling pathway in the melanoma model. However, it would have benefitted from data done with human melanoma cell lines or material to provide a translational aspect.
Please find my specific comments in the following section:
Major points:
1) There are no Figure legends for all supplementary figures. It is possible that they are included in a file named “~$rvin et al IJMS.docx” in the Supplementary folder, but this is not accessible.
2) Supplementary Figure 3: Why is the ALK5 expression so heterogenous (and not visible in many samples)? This should be clarified.
3) The authors use “hepatic replacement” as a proxy for tumor growth in several figures. Please provide an explanation how hepatic replacement was measured and calculated.
4) In Figure 4E it seems that 1D11 leads to an increase of hepatic outgrowth in the Dox-free condition. Is this significant? This should be discussed. Furthermore, the comparative metastatic size of the tumors would be useful (like previously shown in Figure 3D).
5) What is the difference between Figure 5 and Supplementary Figure 4? Also, the differences in Clef4f and CCR2 tumor expression between untreated and Dox-treated animals is significant in one, but not the other figure. Their relevance is unclear to me.
6) Line 471: “However, the latter results were not observed in a second 471 in vivo experiment, in which the sizes of the metastases were larger (Sup. Figure 6 A-C).“
There is no Supplementary Figure 6.
Minor points:
1) The authors need to indicate whether the TGF-b used is of human or of mouse origin. Also, I would recommend a western blot that shows murine TGFBR1 protein expression in comparison to the overexpressed human variant.
2) Some figures seem to be distorted and should be adapted accordingly (e.g. Figures 1A, 2A). Please make also sure that the labels are arranged correctly (e.g. in Figure 1B)
3) Figure 1B: It would be useful to indicate, e.g. in the figure legends, that PAI-1 is the protein product of the Serpine1 gene.
4) The authors mention FACS sorting in their methods. Where in the manuscript was this applied?
5) Line 366: "We found that, upon dox treatment, B16F10 cells expressing caALK5 showed an increased hepatic outgrowth compared to their untreated control (Figure 1B, C)."
should be Figure 3B, C
6) Line 377: “…increased PAI-1 levels were observed in sections of 377 metastatic lesions in caALK5 expressing B16F10 cells (Figure G, H).”
should be “Figure 3G, H”
7) Supplementary Figure 4 should be corrected (has A,B,C, D, F, G, but no E, with corresponding mistakes in the Results referring to this figure).
8) Line 456: “When staining B16F10 control and caALK5 ex- 456 pressing liver metastasis for the CAF marker platelet-derived growth factor receptor 457 (PDGFR)-b, no changes IN CAF staining was observed (Figure 5A, Sup. Figure 4A)”.
9) Line 593: “Dedifferentiation of melanoma cells, resulting in depigmentation and migration of the tumor cells, is known to be promoted BY TGF-b in vitro and in vivo.”
10) Line 511: “With this data, a gene signature was created for caALK5 expressing B16F10 cells (Supplementary Table 3). Interestingly, limited immune suppressive cytokines were found enriched upon caALK5 expression, while reduced number of T cells were found in vivo.“
I believe that the authors created a protein signature (not a gene signature), given that they used mass spec data as source. This should be clarified, also in the Materials and Methods section “Gene set enrichment analysis”. Please indicate examples for possible suppressive cytokines and be more specific about the matrix-associated processes.
Author Response
We thank this reviewer for their kind words and for reading our manuscript thoroughly and providing us with insightful and useful comments. We have considered utilizing human cells for translational aspect. However, human cells can only be transplanted in mice without/altered immune system, making it more difficult to assess our finding on the immune system. We have had our manuscript proofread by MDPI (see certificate in uploaded files). In addition, we have carefully gone through the manuscript again and corrected spelling mistakes.
Below we address the reviewer’s concerns one by one. We hope that the revised manuscript satisfies all the concerns of the reviewer.
Major points:
- There are no Figure legends for all supplementary figures. It is possible that they are included in a file named “~$rvin et al IJMS.docx” in the Supplementary folder, but this is not accessible.
We are sorry the reviewer was not able to assess the supplementary figure legends. We have now provided the supplementary figure legends together with the figures.
- Supplementary Figure 3: Why is the ALK5 expression so heterogenous (and not visible in many samples)? This should be clarified.
In panel A, each lane is a separate clonal cell line, which has been transduced with caALK5 lentivirus. Each infected cell might have a different number of caALK5 integrations (at different genomic locations), leading to differences in expression levels. As such, a single clone was picked (red arrow) showing good, but not too high expression, and this clone was used for further experiments.
In panel B, we can have the same reasoning as for panel A, but then cells were transduced with kiALK5.
In panel C, protein samples from mouse livers containing metastasis are shown. Liver samples for these protein samples were dissected from livers containing variable degrees of metastasis. We were not able to correct for metastasis content in these samples, only total protein content. The variable metastasis content in these samples is underlying the variability in panel C.
In line 139-140, we have included the following sentence: ‘ The liver samples were not corrected for contribution of liver metastasis per sample’, and added this sentence in the results to clarify: The induced expression of ALK5 could be detected at the protein level, although levels varied, possibly due to differential contributions of metastasis in protein samples. (Sup. Figure 3C)’ (line 409-410) and included this in the figure legends of Sup. Figure 3C: ‘Protein samples were not corrected for B16F10 metastasis contribution per sample.’
In line 404-405, we have included the following sentence: ‘Clonal cell lines were selected based on similar caALK5 and kiALK5 expression (Sup. Figure 3A and B).’
We hope this addresses the reviewer’s concerns.
- The authors use “hepatic replacement” as a proxy for tumor growth in several figures. Please provide an explanation how hepatic replacement was measured and calculated.
Our apologies. This was indeed not well described in our manuscript. We have now added a description to the methods section “immunofluorescence”
4) In Figure 4E it seems that 1D11 leads to an increase of hepatic outgrowth in the Dox-free condition. Is this significant? This should be discussed. Furthermore, the comparative metastatic size of the tumors would be useful (like previously shown in Figure 3D).
We thank the reviewer for this observation. We also noticed this trend of increased metastatic outgrowth in liver by 1D11 treatment in the no dox group. This effect was not significant in a two-way ANOVA (P value 0.48). We currently have no explanation for this trend. The relevance of this increase will have to be confirmed in a repeat in vivo experiment or by using another cell line, and by using an additional method of blocking TGF-b signaling in vivo, for example by small molecule TbRI kinase inhibitor treatment. We can only speculate why we see this slight increase upon 1D11 treatment. Since we observe that TGF-b activation (through caALK5 expression) in tumor cells can increase liver metastatic outgrowth of B16F10 cells, the increased outgrowth upon inhibition of TGF-b signaling by 1D11 could be due to effects of ID11 on TGF-b-induced effects on tumor microenvironment. Further studies are needed to validate and possibly explain this result.
To answer the second concern raised by the reviewer, the metastasis from the experiment shown in Figure 4 and Sup. Figure 5 were larger than the metastasis from the experiment shown in Figure 2 and Sup. Figure 4. Because of the larger metastasis, we were not able to determine individual metastasis based on the H2B-mTurquoise expression of B16F10 cells Therefore we could not reliably determine metastatic sizes in this experiment. We did not see a clear difference in number or size of metastasis in these groups by eye. We hope this answers the reviewer’s concern.
5) What is the difference between Figure 5 and Supplementary Figure 4? Also, the differences in Clef4f and CCR2 tumor expression between untreated and Dox-treated animals is significant in one, but not the other figure. Their relevance is unclear to me.
We have utilized different methods of presenting the data. In Figure 5, each dot represents one metastasis. In supplementary Figure 4, all metastases of one mouse have been averaged into one measurement (dot). As this reviewer indicated, the differences disappear if all metastases are averaged. We feel this is not a correct interpretation of the results, as large (size) differences between the metastases were observed. Moreover, the data of the second in vivo experiment already suggest that larger metastases have a different outcome. As such, we felt that including all the individual metastases is a better way of analysis. To be transparent, we included the graphs in which all metastases are combined (Sup Figure 4).
We have now clarified the differences between the figures in the legend of Supplementary Figure 4: ‘This figure shows the average value of all metastasis per liver as shown in Figure 4.’
Similarly, this is also indicated in Sup. Figure 5, where the all values and averages are shown in the same Figure: ‘In the left panels, the average is shown per liver section from n=5 (no dox), n=6 (dox), and n=6 (dox–no dox). In the right panels, all metastases per group are shown.’
6) Line 471: “However, the latter results were not observed in a second 471 in vivo experiment, in which the sizes of the metastases were larger (Sup. Figure 6 A-C).“
There is no Supplementary Figure 6.
Our apologies. We have now corrected this in the text to sup. Figure 5.
Minor points:
1) The authors need to indicate whether the TGF-b used is of human or of mouse origin. Also, I would recommend a western blot that shows murine TGFBR1 protein expression in comparison to the overexpressed human variant.
In this manuscript, we have used TGF-b3 of human origin, which we now indicated in the methods (line 90). As human and mouse TGF-b3 are very conserved and have almost identical amino acid sequence. There is no difference between human and mouse TGF-b3 on murine cells.
The antibody used in e.g. Figure 1B, cross reacts with mouse ALK5 as well. The samples that were not treated with doxycycline show basal murine level of ALK5, which is not detectable on western blot. In line with these observations, mRNA levels of mouse ALK5, shown in supplementary Figure 1B, is also very low. On mRNA levels, we did not observe consistent changes in mouse ALK5 upon TGF-b stimulation or induction of human ALK5 expression. We hope this addresses the reviewers question.
2) Some figures seem to be distorted and should be adapted accordingly (e.g. Figures 1A, 2A). Please make also sure that the labels are arranged correctly (e.g. in Figure 1B)
We thank the reviewer for pointing this out, we have corrected the figures.
3) Figure 1B: It would be useful to indicate, e.g. in the figure legends, that PAI-1 is the protein product of the Serpine1 gene.
We have now mentioned this in the manuscript (line 306).
4) The authors mention FACS sorting in their methods. Where in the manuscript was this applied?
FACS sorting was used to sort for high fluorescence in the H2B-mTurquoise expressing B16F10 cells, necessary to identify the B16F10 cells in the liver sections, as well as to FACS sort these cells into single cells for clonal cell line generation. We have mentioned this now more clearly in the methods.
5) Line 366: "We found that, upon dox treatment, B16F10 cells expressing caALK5 showed an increased hepatic outgrowth compared to their untreated control (Figure 1B, C)."
à should be Figure 3B, C
Thank you, we have changed the reference to Fig. 3B
6) Line 377: “…increased PAI-1 levels were observed in sections of 377 metastatic lesions in caALK5 expressing B16F10 cells (Figure G, H).”
à should be “Figure 3G, H”
Thank you, we have changed the reference to Fig. 3G, H
7) Supplementary Figure 4 should be corrected (has A,B,C, D, F, G, but no E, with corresponding mistakes in the Results referring to this figure).
Thank you, we have corrected this mistake.
8) Line 456: “When staining B16F10 control and caALK5 ex- 456 pressing liver metastasis for the CAF marker platelet-derived growth factor receptor 457 (PDGFR)-b, no changes IN CAF staining was observed (Figure 5A, Sup. Figure 4A)”.
Thank you, we have corrected this mistake.
9) Line 593: “Dedifferentiation of melanoma cells, resulting in depigmentation and migration of the tumor cells, is known to be promoted BY TGF-b in vitro and in vivo.”
Thank you, we have corrected this mistake.
10) Line 511: “With this data, a gene signature was created for caALK5 expressing B16F10 cells (Supplementary Table 3). Interestingly, limited immune suppressive cytokines were found enriched upon caALK5 expression, while reduced number of T cells were found in vivo.“
I believe that the authors created a protein signature (not a gene signature), given that they used mass spec data as source. This should be clarified, also in the Materials and Methods section “Gene set enrichment analysis”. Please indicate examples for possible suppressive cytokines and be more specific about the matrix-associated processes.
We agree with the reviewer that our signature is a protein signature. The protein names in this signature were converted to gene names to compare our signature with publicly available gene expression data, which caused the name shift from protein to gene signature. We have corrected this to protein signature in the manuscript, and included under section “Gene set enrichment analysis” that protein names were translated into gene names in order to compare this signature with gene expression datasets: ‘Protein names from the protein signature identified with the secretome analysis were converted to gene names.’ (line 277-278).
The scope of our manuscript ends with our presented (protein) signature. We touch upon potential roles of Periostin, CTGF and PAI-1 in our discussion. However, as we have not investigated these roles or the roles of other secreted proteins, we can only speculate about these effects and which secreted protein, or a combination of multiple proteins, contributes to the effect we observe. Additional in vivo experiments with neutralizing antibodies for these secreted proteins, or knockout B16F10 cell lines for genes of these proteins, will be necessary as follow-up experiments. As we did not do any follow-up experiments on specific proteins in the secretome, or (matrix-associated) processes, we decided to present the data as a protein signature associated with increased melanoma liver metastatic outgrowth upon TGF-b pathway activation. As the gene set enrichment analysis revealed an increase in processes regarding extracellular matrix remodeling, we highlighted these global processes. To further explain the role of specific cytokines would be cherry picking and speculation, which we feel is not appropriate at this stage. As such, we have adapted the text slightly to indicate that matrix could be remodeled, which could lead to a physical barrier which might limit the penetration of T cells. We also indicate that further studies are required to investigate the underlying mechanisms in more depth.
In the discussion, we have now rephrased this sentence:’ Analysis of enrichment of pathway and biological processes showed no direct evidence for immune suppressive processes upon caALK5 expression (Figure 6B), while reduced number of T cells were found in vivo’ , line 613-616.
(…)
‘ However, future studies are needed to investigate the underlying mechanisms in more depth. ‘ (620-621)
Round 2
Reviewer 1 Report
This paper has been revised in a satisfactory manner and is now suitable for publication.
Reviewer 2 Report
The authors have addressed all issues, and I have no further comments.